# The Efficacy of Animal-Assisted Therapy in Patients with Dual Diagnosis: Schizophrenia and Addiction

**DOI:** 10.3390/ijerph19116695

**Published:** 2022-05-30

**Authors:** Miguel Monfort, Ana Benito, Gonzalo Haro, Alejandro Fuertes-Saiz, Monserrat Cañabate, Abel Baquero

**Affiliations:** 1TXP Research Group, Universidad Cardenal Herrera-CEU, CEU Universities, 12006 Castellón de la Plana, Spain; mmmontolio@yahoo.es (M.M.); anabenitodel@hotmail.com (A.B.); gonzalo.haro@uchceu.es (G.H.); alejandro.fuertessaiz@uchceu.es (A.F.-S.); montserrat.canabate@uchceu.es (M.C.); 2Proyecto Amigó, 12006 Castelló de la Plana, Spain; 3Torrente Mental Health Unit, Hospital General Universitario Valencia, 46014 Valencia, Spain; 4Severe Dual Disorder Program, Mental Health Department, Hospital Provincial de Castellón, 12002 Castelló de la Plana, Spain; 5Psychiatry Service, Hospital La Salud, 46021 Valencia, Spain; 6University Clinic Hospital, 46014 Valencia, Spain

**Keywords:** animal-assisted therapy, addiction, residential treatment, daily living skills, dual diagnosis

## Abstract

The objective of the study was to evaluate the efficacy of an animal-assisted-therapy (AAT) program in patients diagnosed with schizophrenia-spectrum disorders and substance-use disorders in residential treatment in order to intervene in the remission of negative and positive symptoms and improve quality of life and adherence to treatment, favouring the clinical stabilization of patients who participate in the AAT program, within the context of a mental-illness-treatment device. This was a quasi-experimental prospective study with intersubject and intrasubject factors. The sample comprised 36 patients (21 in the experimental group and 15 in the control group) who were evaluated at three time points (in the 3rd, 6th, and 10th sessions). The program lasted 3 months and consisted of 10 sessions that were implemented once a week, with a maximum participation of 10 patients per group. The participants were evaluated with the *Positive and Negative Syndrome Scale* (PANSS) for schizophrenia and the *Life Skills Profile-20* (LSP-20) questionnaire. We observed a decrease in the positive symptoms of psychosis (*F:* 27.80, *p* = 0.001) and an improvement in functionality (*F:* 26.70, *p* < 0.001) as the sessions progressed. On the basis of these results, we concluded that AAT seems to be valid as a coadjuvant therapy as part of the rehabilitation processes of patients diagnosed with schizophrenia and addiction-spectrum disorders (dual diagnosis).

## 1. Introduction

Schizophrenia is considered one of the most challenging mental disorders known to science due to the complexity of the disease and the presentation of the factors that comprise it. The National Institute of Mental Health [1] defines schizophrenia as a chronic, inflexible, and disabling mental illness, in which contact with reality and basic psychological processes are lost. In addition to the positive and negative symptoms of the illness, the involvement of other spheres, such as the cognitive and affective domains, tends to limit the functionality of patients in all areas of their lives (work, family, and leisure, among others). In this context, the negative symptoms (emotional blockage or the inability to express affection) and the positive symptoms (hallucinations, delusions, stereotypies, catatonia, and disorganisation of thought, language, and behaviour) sometimes disable some personal faculties, hindering emotional expression and the course and performance of thought. All this symptomatology, in many cases, reduces interpersonal functionality and hinders the favourable evolution of treatment, and therefore reduces functionality [2,3], and even hinders pharmacological treatment [4].

The presence of a substance-use disorder (SUD) and another mental illness in the same patient is called dual diagnosis (DD). DD, this coexistence of a mental disorder with substance use, makes treatment difficult and challenging due to the worsening of the clinical manifestations, as well as generates a worse evolution in care for the consumption disorder, configuring a serious clinical profile that requires continuous and specific attention, with problems such as those due to more frequent relapses, more continuous psychiatric care, etc., and, according to recent studies, DD is very common in schizophrenia, with a prevalence of around 48% [5,6]. Patients with schizophrenia who present a DD tend to have more relapses, a poorer clinical evolution, and more difficulties in the rehabilitation process [7,8]. In addition, schizophrenia does not affect patients equally, and its symptomatologic course is heterogeneous and variable, with this element also determining the therapeutic approach. Schizophrenia is associated with a high degree of incapacitation, a poor functional prognosis at work, and inferior levels of social and daily life skills, all of which notably reduce patient quality of life [9].

Current therapeutic proposals for this disease are multidimensional, but they mainly consist of combining antipsychotic drugs and psychosocial and psychoeducational interventions, prioritising the goal of achieving greater patient autonomy. In this multidimensional context, animal-assisted therapy (AAT) can be applied as a complement to standard treatments, thereby helping to provide patients with the skills required for them to develop everyday life and social functionality [10,11,12,13,14]. AAT is defined as an intervention with a specific, structured, and directed goal, which is implemented by professionals in the field of health and education, and which uses an animal as a therapeutic instrument and as another tool in the rehabilitation process. AAT can be implemented by using different animals, including dogs, cats, and horses, among others. However, dogs are most often used in this type of therapy because of the variety of breeds available, their ease of training, and their commonplace use as positive reinforcements for patients. This intervention is carried out in different settings, such as schools, nursing homes, prisons, or mental-health-treatment resources [15,16,17,18].

Patients who are diagnosed with a schizophrenia-spectrum disorder and substance use have impaired levels of activity and social functioning, and reduced social problem-solving strategies, which can be the object of intervention from the AAT. Patients benefit from this therapy through the animal interaction. These benefits include socialisation, stress, and anxiety management, and the enhancement of basic functional skills, which all favour their motivation and further participation in treatments [15,16,19,20,21]. However, AAT has not yet been evaluated as a potential treatment in patients with DD. Therefore, the objective of the current study was to assess the efficacy of the AAT program in terms of the evolution of the psychotic symptomatology and functionality of patients with schizophrenia-spectrum disorder and SUD. We hypothesised that: (1) AAT decreases positive and negative psychotic symptoms and general psychopathology; (2) AAT increases functionality; and (3) the positive and negative symptoms mediate how AAT affects functionality.

## 2. Materials and Methods

This was a quasi-experimental prospective study in patients diagnosed with schizophrenia-spectrum disorders and SUD in residential treatment, and, specifically, the detoxification program in the Therapeutic Community of Project Amigó Castellón. The patients included in the research were selected through nonprobabilistic nonrandomised consecutive sampling over 24 months. The patients were sequentially allocated to the experimental group (EG) or the control group (CG), in that order. The definitive sample size was set at 36 patients, of which 21 took part in the AAT program, and the remaining 15 did not. Thus, the study presents an intersubject factor according to participation in the EG or CG. The inclusion criteria were the following: both sexes over 18 years old diagnosed with schizophrenia-spectrum disorder and DD, cooperating with treatment in a residential resource in the province of Castellón de la Plana, having been prescribed psychopharmacological treatment, and the willingness to participate in the study and having signed the informed consent.

The exclusion criteria were the following: the presence of a mental disorder other than the object of study, individuals whose functioning may be impaired by factors not specifically related to schizophrenia or DD (having received electroconvulsive therapy, severe cognitive impairment, etc.), not following the indications of the pharmacological regimen and the guidelines proposed by the centre’s rehabilitation team, and allergy or phobia to dogs. In the EG, 5 subjects dropped out, and 3 were excluded for not meeting the inclusion criteria in the study. In the CG, 3 participants dropped out, and 2 were not accepted because they did not meet the established criteria. There is no significant difference between both groups in the percentage of excluded and dropouts ($\chi^{2}$ = 0.49; *p* = 0.976).

The CG received the standard treatment provided at the residential treatment centre where the participants were undergoing rehabilitation, which consisted of sessions of psychotherapy, psychoeducation, and a specific program for the treatment of schizophrenia and substance use from a cognitive behavioural approach, while the EG received the standard care and also participated in the AAT as a complementary therapy. The AAT program was implemented in 10 sessions delivered over a maximum of 3 months. One session lasting 45 min was delivered per week in groups of a maximum of 10 patients. A therapy dog, social educator, psychologist, AAT technician, and dog trainer all participated in delivering the sessions. Each therapy session starts with the presentation and greeting to the patients, animal, and therapist, who make up the therapy group. The greeting consists of introducing yourself to the dog and petting it at the beginning of the session. The warm-up exercise continues, with a very specific objective: focus the attention of the patient and the animal on the initiated group. The exercises are simple: introduction, greeting, brushing, and caressing specific areas. After the warm-up, the specific exercise of the session is given, and dog–patient occurs to achieve the objectives set for the session. Before the end of the session, a time of sharing experiences, sensations, and emotions is shared, and the session concludes with the farewell of the dog and therapist patients, and a call for the next session.

In the session, patients are informed of different aspects of canine behaviour, such as their care, feeding, and the learning or extinction of behaviours through positive reinforcement (prizes in the form of food, caresses, or congratulations), and the behaviour that should be observed.

The materials used in the sessions were food prizes, clicker, drinker, leash, bib, brush, mitten, comb, balls, cones, rings, cardboard, envelopes, blackboard, backpack, and table.

The intrasubject factor was the AAT session number (3rd, 6th, and 10th sessions), when two different questionnaires were completed: The *Positive and Negative Syndrome Scale* (PANSS) for schizophrenia (comprising three factors: positive symptoms, negative symptoms, and general psychopathology). The validity and reliability of the scale has been demonstrated in people with schizophrenia in Spain. The internal consistency of the positive scale is moderate (Cronbach’s α = 0.62), high for the negative scale (α = 0.92), and modest for general psychopathology (α = 0.55). Interobserver reliability is good for the positive and negative scales (ICC = 0.71 and 0.80, respectively), and moderate for the general psychopathology scale (ICC = 0.56). Construct validity between positive and negative scales is good ((r = 0.09), n.s., between the positive and negative scales). The criterion validity is high, showing a high correlation of the positive and negative scales with the SAPS and SANS scales (r = 0.70 and r = 0.81, respectively) [22,23]. The *Life Skills Profile-20* (LSP-20). The scale shows an internal consistency of 0.85, which ranges from 0.69 to 0.79 for the five areas assessed. Interobserver reliability is above 0.60 for all five areas [24,25]. To enhance the validity of the data, different people performed the therapy, data collection, and analysis. The person collecting the data was blind to the group to which the subjects belonged and was trained in the use of the evaluation instruments. To standardize the results, we use the PANSS percentile scores.

### 2.1. Statistical Analysis

Data analysis was performed with SPSS software for Microsoft (version 23.0; IBM Corp., Armonk, NY, USA). After the exploratory and descriptive analyses, quantitative descriptive variables were compared using Student *t*-tests (checking normality with the Shapiro–Wilk test and equality of variances with the Levene test), and categorical variables were compared using Pearson chi-squared tests. The dependent variables were subsequently compared between the groups and at the three time points by applying mixed MANOVA tests by using the group as the intersubject factor, and the evaluation time point as the intrasubject factor. Bonferroni correction for multiple comparisons was used. Finally, the data were modelled using PROCESS v3.4 for SPSS [26] to test our three hypotheses.

### 2.2. Ethical Factors

This study was designed in accordance with the principles set out in the 2013 Declaration of Helsinki, Spanish Organic Law 3/2018, of December 5 on the Protection of Personal Data, and the law on the Guarantee of Digital Rights and Regulation (EU) 2016/679 of the European Parliament, and of the European Council of April 27, 2016, on Data Protection (GDPR). All patients who participated in the study signed their informed consent.

In relation to the therapy dog, all the criteria established in the following laws were met: (1) the IAHAIO White Paper of 2018; (2) the European Convention on the protection of companion animals of 1987; (3) the Declaration on Animals, approved on 15 October 1987; (4) the Valencian regional government (Generalitat) Law 12/2003 of April 10 on Assistance Dogs for People with Disabilities (DOGV № 4479, of 4 November 2003); (5) Decree 167/2006 of November 3 by the local Valencian council that developed Law 12/2003 of April 10 of the Generalitat on Assistance Dogs for People with Disabilities (DOGV № 5382 of 7 November 2006); and (6) the Order of 30 May 2007 of the Ministry of Social Welfare that approves the application model for the procedure used to recognise assistance dogs for people with disabilities, the minimum content of pet-assisted therapy projects (TAAC in its original initialism in Spanish; DOCV № 5532, of 6 December 2007), and the pertinent documents required to request assistance-dog recognition. This study has the report of the Ethics Committee for Biomedical Research of the CEU Cardenal Herrera University, registry CEEI21/197, and the relevant registration in the clinical trials, with registry NCT05103865.

## 3. Results

### 3.1. Sociodemographic and Clinical Characteristics

No significant differences were found between the EG and the CG in terms of age, sex, marital status, family relationships, or employment status.

The mean age of the sample was 40.3 years (*SD* = 6.1), with the patients having received a mean of 2.18 treatments (*SD* = 0.81), having made a mean 0.65 suicide attempts (*SD* = 0.72), and with a mean of 1.3 hospital admissions to emergency mental health services (*SD* = 1.52). Most of the patients (86.1%) were male. The most prevalent marital status was single (72.2%; *n* = 26), followed by separated or divorced (25%; *n* = 9), and only 2.8% (*n* = 1) of the participants were married; most of them maintained contact with their families (66.7%; *n* = 24). See Table 1.

In terms of employment, half of the participants were employed and on temporary sick leave, and the other half were unemployed or were receiving some type of social benefit. Regarding the education of the patients, we found that 2.8% (*n* = 1) had not completed any studies; 19.4% (*n* = 7) had started but not completed their primary education; 8.3% had graduated from school with a basic education; 61.1% (*n* = 22) had completed the Unified Polyvalent Baccalaureate, compulsory secondary education, or level 1 vocational training; 5.6% (*n* = 2) had higher vocational training; and only 2.8% (*n* = 1) of the sample had completed university-level studies.

Regarding substance use, no significant differences were found between the CG and EG, and so they were considered comparable. The participants had been using substances for a mean of 25.2 years (*SD* = 6.6). Polydrug use of tetrahydrocannabinol (THC) + cocaine + alcohol stood out, with a prevalence of 44.4% (*n* = 16) in the sample groups. In the EG, the second most consumed substance was alcohol (16.7%; (*n* = 6)), followed by cocaine (11.1%%; (*n* = 4)), heroin + cocaine (5.6%; (*n* = 2)), and, finally, THC (2.8%%; (*n* = 1)). In the CG, the polydrug use of THC + cocaine + alcohol was followed by the use of cocaine, alcohol, and heroin + cocaine (5.6%; *n* = 2 for all of them), and lastly, THC (2.8%; *n* = 1).

In relation to comorbid medical pathologies, no differences were found between the groups, which highlights the fact that the majority (63.9%; *n* = 23) did not present any organic pathologies. However, 5.6% (*n* = 2) had hepatitis B and C (HBV + HCV) coinfection, while 8.3% had human HIV + HBV, 11.1% had human immunodeficiency virus (HIV) + HCV, and 5.6% had a multiple HIV + HBV + HCV infection. Regarding the psychopharmacological treatment, there were no statistically significant differences between the groups. Of note, after treatment with antipsychotics, the most used drug in the CG was antidepressants (33.3%; *n* = 5), followed by anxiolytics (13.3%; *n* = 2), which was similar to the EG, in which 19% (*n* = 4) of the patients used antidepressants, and 28.6% (*n* = 6) used anxiolytics.

### 3.2. Evaluation of Positive and Negative Psychotic Symptomatology and General Psychopathology

The MANOVA analysis found significant effects for the group (*F*: 4.799; *p*: 0.004; ${\hat{\eta }}^{2}$: 0.382; (1 − β): 0.920), the measurement time point (*F*: 14.266; *p* < 0.001; ${\hat{\eta }}^{2}$: 0.809; (1 − β): 1), and the interaction between both these factors (*F*: 10.52; *p* < 0.001; ${\hat{\eta }}^{2}$: 0.757; (1 − β): 1). As shown in Table 2 and Figure 1a,d, compared to the CG that only received standard treatment, the results for the EG indicate that AAT produced a reduction in positive symptoms and an increase in the skills required for everyday living in patients with schizophrenia-spectrum disorders who also presented SUDs.

The positive symptoms in both groups were comparable at the beginning of the study and decreased throughout the treatment period, although they decreased significantly more in the EG, which presented fewer positive symptoms at time points 2 (*F*: 25.66; *p*: 0.006) and 3 (*F*: 27.81; *p*: 0.002), compared to the CG (Figure 1a). Regarding the negative symptoms (Figure 1b), there were significant differences between the groups at the beginning of the study (*F*: −19.14; *p*: 0.030), with the EG presenting a higher score, and the groups were not comparable in this variable. Even so, this difference was no longer present by the end of the study, and so we inferred that the score in the EG had reduced more than the CG score. The levels of general psychopathology (Figure 1c) decreased in both groups, with no significant differences between them throughout the study. The daily life skills (Figure 1d) of both groups were comparable at the beginning of the study and increased thereafter, but there was significantly more improvement in the EG, which presented more skills at the end of the AAT compared to the CG (*F*: −20.44; *p*: 0.001).

Figure 2 shows the mediation model that was used to evaluate the relationships between the variables studied. However, the results regarding the negative symptoms were not reliable because the two groups were not comparable at the beginning of the study. As shown, the AAT was related to a decrease in the positive symptoms and an increase in functionality, without the changes in psychotic symptoms appearing to mediate the effect of AAT on functionality.

## 4. Discussion

To the best of our knowledge, this is the first scientific article to have analysed the response of patients with a DD (schizophrenia and SUD) to treatment with AAT in terms of their psychotic symptomatology and daily functionality. On the basis of the results of this study, there was some evidence that our first hypothesis was partially fulfilled because AAT reduced the positive symptomatology of the patients in the EG. In addition, AAT may have also helped to reduce the negative symptomatology; however, we cannot conclude the latter because the results were likely masked by the baseline differences in both groups at the beginning of the intervention. With regard to the second hypothesis, this work confirms that AAT helped improve the daily life skills of patients with schizophrenia and SUD, thus improving their everyday functionality. However, the third hypothesis was rejected because the positive and negative symptoms did not seem to mediate how AAT affected the functionality of the patients studied.

With regard to the positive psychotic symptomatology, our results were consistent with those of other authors that have independently studied the response of patients with schizophrenia or SUD to AAT. There is a lot of heterogeneity in the few studies that have evaluated the efficacy of AAT on the positive symptoms of schizophrenia. A systematic review by Hawkins et al. (2019) included seven randomised placebo-controlled clinical trials, among which only one found a significant reduction in the positive and emotional symptomatology of patients with schizophrenia [27]. A few studies have also analysed the presence and evolution of psychotic symptoms in patients with SUD after completing an AAT intervention. Contalbrigo et al. (2017) published a study that carried out an AAT intervention in men with SUD over a period of 6 months and, in line with our results, they observed a significant reduction in psychoticism and paranoid ideation [28].

Other work has previously shown how the relationship between dogs and humans improves the social support perceived by patients and facilitates the learning of effective coping strategies [29]. These two factors, in turn, decrease levels of psychological stress, which can be measured through a decrease in cortisol levels, heart rate, and blood pressure [30]. An important role for basal cortisol levels in patients at risk of psychosis has recently been demonstrated, and it has been determined that this factor is directly related to the appearance of psychotic symptoms [31]. This current work demonstrates, for the first time, how an AAT intervention can reduce positive symptoms in patients with a DD of schizophrenia and SUD, and it adds to the body of evidence derived from previously published studies on the role that this type of therapy could play in the future treatment of patients with schizophrenia and/or SUDs.

With regard to the negative symptomatology, it is worth noting that these symptoms are usually refractory to treatment with antipsychotics, and they notably affect the quality of life and functionality of these patients [32]. In the review by Hawkins et al., only one study found a significant reduction in the negative symptomatology in patients with schizophrenia after an AAT intervention [27]. From a neurobiological point of view, some mechanisms of action have been hypothesised that could explain why we would expect AAT to reduce negative symptoms in patients with schizophrenia. One such mechanism, postulated by Hawkins et al. (2019), is that of oxytocin, a hormone whose intranasal administration has been shown to decrease the levels of psychotic symptomatology in patients with schizophrenia [33], and whose levels increase with the interaction of human beings with dogs [34]. Our own results show how the significant differences in negative symptomatology in the EG and CG at the beginning of the treatment had disappeared by the end of the intervention, apparently as the result of a decrease in the intensity of these symptoms in the treatment group. However, although our findings point in this direction, we cannot conclude that AAT reduced the negative symptoms in our EG of patients. This difference in baseline psychopathology may have also masked the detection of significant differences in the general psychopathology of the patients we studied.

In this work, we were also able to objectify how the daily functioning of patients with a DD (schizophrenia and SUD) treated with AAT significantly improved compared to the CG. According to our results, this was a direct effect of AAT, whereby the improvement in the psychotic symptoms did not function as a mediator. Of note, the effect of AAT on daily functioning has previously been studied in different clinical trials, and especially in patients with schizophrenia. For instance, Barak et al. (2001) studied adaptive social functioning and the activities of daily living in 20 patients diagnosed with schizophrenia. The group receiving the AAT showed increased levels of mobility, interpersonal contact, communication, and activities of daily living, including in relation to personal hygiene and self-care.

However, contrary to our study, most of the study cohort in the aforementioned study were women [35]. Another study, also conducted in a sample of patients with schizophrenia, identified an improvement in the profile of daily living skills and an increase in the social-interaction score [36]. Moreover, an improvement in appropriate social behaviours, greater interaction between patients, and an increase in smiles and satisfaction were also identified in a study on patients with both schizophrenia and a history of substance abuse in the group that received an AAT intervention, compared to the controls [37]. In addition, Uhlmann et al. (2019) analysed a sample of patients with SUD and concluded that patients perceived a more pleasant atmosphere when dogs were present, and felt that they had an opportunity to modify their dysfunctional behaviour patterns by acquiring new personal, social, and emotional competencies [38].

Thus, all these results, including those presented in this present study, invite us to think of AAT as a directly effective therapy for promoting and improving the activities of daily living, regardless of whether it be in patients with or without psychotic symptoms, in those diagnosed exclusively with schizophrenia, or in patients presenting a dual phenotype. Within the policies and actions in mental health, it seems appropriate to indicate that this type of intervention must be considered, since they follow the guidelines of the current mental health clinical plans. However, before concluding, it is important to mention the limitations of this work. First of all, this study is considered quasi-experimental because the groups were not randomized, which may have influenced our results, and so they should be interpreted with caution. In addition, we did not consider the gender variable, which may have influenced the results [39,40], although there were no significant differences between the groups for this variable at the start of treatment. Another important limitation is that, due to difficulties with the start of the study, the subjects were not evaluated at baseline, but instead began to be evaluated in the third session. Moreover, as the AAT was not compared to another complementary therapy, extra time spent in therapy by the AAT group may have hindered our results. The small sample size and the inequality between the groups would make it convenient to use a nonparametric test; however, the nonparametric analysis of the interaction is not incorporated into the statistical packages [41], and so MANOVA was used. Another problem with MANOVA is that, if a variable has been entered as within subjects, it is not possible to also enter it as a covariate, which would have been very useful with the negative PANSS score, since the two groups start at different points. Another important issue is that, although exclusions and dropouts occurred equally in both groups, it is not possible to disentangle the effect of the AAT from the clinical characteristics at the time of dropout or refusal to participate, which may influence the outcome averages, as it is a small sample. Finally, this study was conducted in a specific therapeutic scenario, which makes it difficult to extrapolate the results to other care resources, such as specific centres for people with mental disorders (CEEMs in their Spanish acronym), day hospitals, etc. [42,43,44]. Nor did we consider the possible underlying animosity or affinity patients may have for these animals, which could have altered the effectiveness of the AAT.

## 5. Conclusions

In summary, this is the first study to evaluate the efficacy of AAT on the psychotic symptomatology and everyday functionality of patients with DD (schizophrenia and SUD). The implementation of AAT as a complementary approach to the standard therapy provided at the residential centre where this study was conducted reduced the positive symptoms and improved the daily functionality of patients with DD. In addition, we demonstrated that AAT directly affects the daily functioning of patients independently of psychotic symptomatology. Therefore, this work adds to the body of evidence for the efficacy of AAT in patients with a DD as one of the most vulnerable patient groups.

## Figures and Tables

**Figure 1 ijerph-19-06695-f001:**
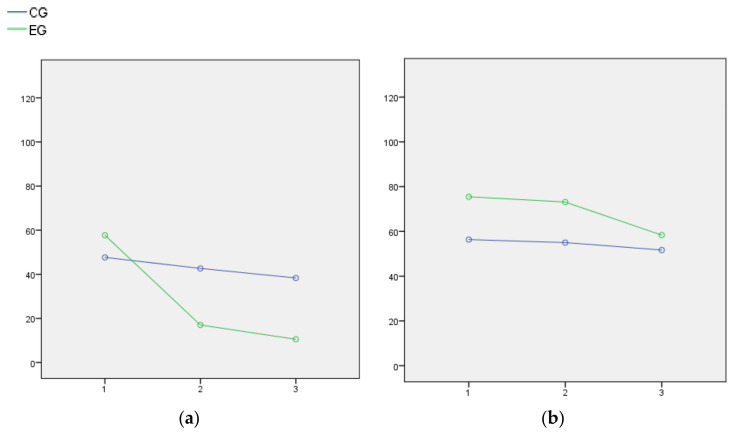
Evolution of the *Positive and Negative Syndrome Scale* (PANSS) percentile scores and *Life Skills Profile-20* (LSP-20) results during the implementation of animal-assisted therapy (AAT), compared to a control group who did not receive AAT: (**a**) PANSS positive; (**b**) PANSS negative; (**c**) PANSS general psychopathology; (**d**) LSP-20. Note: Time = results measurement session number: 1 after the 3rd session of AAT; 2 after the 6th session of AAT; 3 after the 10th session of AAT. CG: control group in blue; EG: experimental group in green.

**Figure 2 ijerph-19-06695-f002:**
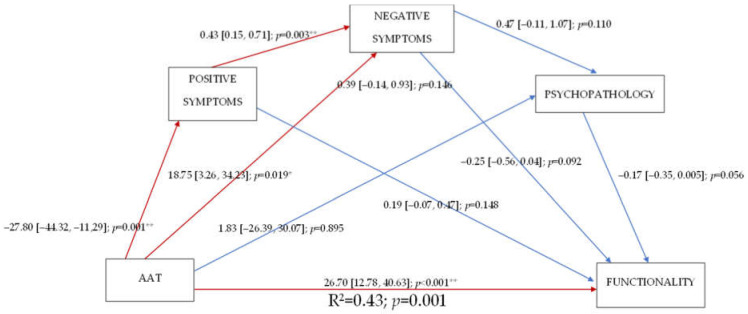
A mediation model for the effect of animal-assisted therapy (AAT) on the positive and negative symptoms of schizophrenia. Note: AAT: animal-assisted therapy. * *p* < 0.005. ** *p* < 0.001.

**Table 1 ijerph-19-06695-t001:** Main sociodemographic characteristics.

Sociodemographic	AAT vs. No AAT	*n*	Mean	Standard Deviation
Age	CG	15	39.53	5.20
	EG	21	41.85	6.86
Number of treatments	CG	15	2.06	0.79
	EG	21	2.28	0.84
Suicide attempts	CG	15	0.73	0.79
	EG	21	0.57	0.67
Hospital admissions	CG	15	1.06	1.75
	EG	21	1.52	1.36
Years of consumption	CG	15	24.33	5.87
	EG	21	25.95	7.22

Note: CG: control group; EG: experimental group; AAT: animal-assisted therapy.

**Table 2 ijerph-19-06695-t002:** Differences in the *Positive and Negative Syndrome Scale* (PANSS) percentile scores and *Life Skills Profile-20* (LSP-20) scores between the experimental group and the control group.

Test	Time	CG (*n* = 15) Mean *SD*	EG (*n* = 21) Mean *SD*	Mean Difference CG-EG *p*-value	Mean Difference *t* 1 − 2*p* 2 − 3*p* 1 − 3*p*	Group Effect *F* *p*-Value ${\hat{\eta }}^{2}$ (1 − β)	*t* Effect *F* *p*-Value ${\hat{\eta }}^{2}$ (1 − β)	*i* Effect *F* *p*-Value ${\hat{\eta }}^{2}$ (1 − β)
PANSSp	1	47.66 33.53	57.71 19.08	10.04 0.261	22.85 <0.001 **	3.428 0.073 0.092 0.436	47.69 <0.001 ** 0.584 1	23.95 <0.001 ** 0.413 1
	2	42.66 34.99	17.00 17.29	25.66 0.006 **	5.40 0.065			
	3	38.33 36.23	10.52 7.91	27.81 0.002 **	28.26 <0.001 **			
PANSSn	1	56.33 29.90	75.47 20.73	19.14 0.030 *	1.85 1	4.369 0.044 * 0.114 0.528	8.76 <0.001 ** 0.205 0.965	3.085 0.052 0.083 0.577
	2	55.00 28.47	73.09 14.78	18.09 0.018 *	9.04 0.001 **			
	3	51.66 26.56	58.33 17.70	6.66 0.372	10.90 0.016 *			
PANSSpg	1	68.00 32.99	85.00 17.32	17.00 0.052	21.11 <0.001 **	0.125 0.725 0.004 0.064	20.03 <0.001 ** 0.371 1	3.617 0.032 * 0.096 0.650
	2	56.00 33.49	54.76 31.87	1.23 0.911	6.00 0.075			
	3	52.33 35.39	46.42 36.16	5.90 0.629	27.11 <0.001 **			
LSP-20	1	95.86 27.13	80.90 19.71	14.96 0.063	14.37 0.001 **	0.706 0.407 0.020 0.129	23.91 0.001** 0.413 1	15.56 <0.001** 0.314 0.999
	2	98.13 27.24	107.38 14.43	9.24 0.195	7.73 0.003 **			
	3	100.26 23.70	120.71 11.25	20.44 0.001 **	22.10 <0.001 **			

Note: Time = results measurement session number: 1 after the 3rd session of AAT; 2 after the 6th session of AAT; 3 after the 10th session of AAT. CG: control group; EG: experimental group; PANSSp: Positive and Negative Syndrome Scale (PANSS) positive-symptom results; PANSSn: PANSS negative-symptom results; PANSSpg: PANSS general pathology results. LSP-20: Life Skills Profile-20 questionnaire results for daily life skills in real clinical contexts. t: time; I: interaction.Note: * *p* < 0.05, ** *p* < 0.005.

## Data Availability

Not applicable.

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
