# Peer review of "The Efficacy of Animal-Assisted Therapy in Patients with Dual Diagnosis: Schizophrenia and Addiction"

_ijerph, 2022, doi:10.3390/ijerph19116695_

Round 1
Reviewer 1 Report
Dear authors of the article
The topic is of interest and is novel since there are few published works with the same approach. I believe it is a relevant and pertinent work for mental health professionals.
However, I would like to point out some areas of opportunity:
1- The introduction is concrete and contains valuable information, however, I consider that it could be expanded to provide more details on dual diagnosis, as well as on animal-assisted therapy, as these are specialized topics with which most readers might not be familiar.
2- The results of sociodemographics and clinical characteristics could be presented using a table that makes them easier to read.
3- Table 1 is difficult to read and confusing. The format could be improved to make it easier to read and more self-explanatory.
4- The quality of Figure 1 is poor and the footnote is not sufficiently explanatory of the information presented.
5- Figures 1 and 2 do not follow the instructions for inserting figures and diagrams of the journal since you place a title like the tables and a footnote, which is incorrect since for these elements only a figure footnote is requested indicating the number, title and other information.
6- Dear authors, I do not know if it is an error but I notice that you have added the same tables and figures in the appendix as in the body of the article
Author Response
Abel Baquero Escribano
Department of Medicine and Surgery, Universidad Cardenal Herrera-CEU. Castellón
13Th May 2022
International Journal of Environmental Research and Public Health
Dear Revisor 1:
Thank you for your invaluable help in reviewing our paper entitled “The efficacy of animal-assisted therapy in patients with schiz-ophrenia and addiction (dual diagnosis)” by Miguel Monfort, Ana Benito, Gonzalo Haro, Alejandro Fuertes, Monserrat Cañabate and Abel Baquero for consideration by the International Journal of Environmental Research and Public Health.
We confirm that we have followed your instructions, you will be able to check the changes in the forwarded paper, please understand that due to limitations of the study and its context it may not have been possible for us to develop all of them, although we consider that thanks to your invaluable collaboration we have been able to improve our proposal.
We enclose the document in which you kindly helped us, as well as others since we consider that you should have a new global vision with all the modifications. We detail the modifications and detect them in the text:
The topic is of interest and is novel since there are few published works with the same approach. I believe it is a relevant and pertinent work for mental health professionals.
However, I would like to point out some areas of opportunity:
1- The introduction is concrete and contains valuable information, however, I consider that it could be expanded to provide more details on dual diagnosis, as well as on animal-assisted therapy, as these are specialized topics with which most readers might not be familiar.
Response 1: This aspect is modified in the text by expanding the information
2- The results of sociodemographics and clinical characteristics could be presented using a table that makes them easier to read.
Response 1: Modification is made.
3- Table 1 is difficult to read and confusing. The format could be improved to make it easier to read and more self-explanatory.
Response 1: Modification is made.
4- The quality of Figure 1 is poor and the footnote is not sufficiently explanatory of the information presented.
Response 1: Modification is made.
5- Figures 1 and 2 do not follow the instructions for inserting figures and diagrams of the journal since you place a title like the tables and a footnote, which is incorrect since for these elements only a figure footnote is requested indicating the number, title and other information.
Response 1: Modification is made.
6- Dear authors, I do not know if it is an error but I notice that you have added the same tables and figures in the appendix as in the body of the article
Response 1: Modification is made.
Thank you for your consideration of this manuscript.
Sincerely,
Abel Baquero

Reviewer 2 Report
I have reviewed the paper title: "The efficacy of animal-assisted therapy in patients with schizophrenia and addiction (dual diagnosis). The paper is reasonably written that are in line the scope of the journal issue. The relevant and contributes to existing literature. Therefore, I recommend this paper subject to the following the revisions.
- Reorganise the contents of this manuscript to improve the flow and coherence for easy understanding. Where appropriate, insert sub-headings to segregate different points for clear illustration and discussion.
- In the abstract and introduction, clearly specify the research problems and research objectives. These are important components of a manuscript in guiding readers through the research. What’s the keypoint that significant for Schizophrenia patients for this study design. What's paper to clear research gap? please clarify.
- Introduction; You should indicate a gap in the previous research, or extend previous knowledge in some way. The introduction is not clear and does not help the reader in the understanding of the whole paper. The topic investigated is not clarified nor explained in terms of the main results obtained. Moreover, please add an outline of the paper at the end of the introductory section that will help the reader in understanding the complex structure of the paper.
- In the methodology section, expand the methods used in 2 group design, Have the data been validated? how to ascertain the validity of the information used for this research? So, The methodological, material and methods section does not contain any methods at this stage. Please better explain the details regarding the questionnaire (when, where, how are selected firms, which sectors etc.). Please report the psychometric properties of all assessment/questionaires that you used in this research. And brief the detail of your program 10 sessions.
- Suggest including a research framework covering all the relevant variables for clear illustration and easy understanding.
- Include a section on practical implications for possible adoption by the targeted audience. Should there be policy recommendations from this research in mental health and psychiatric fieldwork?
- Ensure in-text citations are done in the correct format. Please refer to manuscript formatting guidelines.
- Table 1, some symbol not clear, please check it.
- Figure 2. I think you can upgrade to high figure standard. Please improve design of illustration and Professional styles of figure.
- The discussion section also need to add compatibility with recent studies and how the results of this studies varied compared to existing studies.
- The policy recommendations required extension in terms of key variables.
- How about the limitation of this study? Please explain.
- The manuscript required native Enlight edit and proofing, The are a few grammatical and sentence structure errors.
- Please confirm the reference styles, as several references in the bibliography show inconsistent formatting. Some references have the complete name of journals while others have abbreviations. Please update and make it consistent.
This article is strongly recommended for publication after incorporating certain changes. This article needs thorough proofreading. Overall quality of Language is good. Just minor grammatical mistakes are found. All tables and figures are relevant. Research Methodology has been well defined. All data are aligned to the findings of the research. This article is good attempt in the field experimental research design and will be beneficial for future researchers.
Author Response
Abel Baquero Escribano
Department of Medicine and Surgery, Universidad Cardenal Herrera-CEU. Castellón
13Th May 2022
International Journal of Environmental Research and Public Health
Dear Revisor 2:
Thank you for your invaluable help in reviewing our paper entitled “The efficacy of animal-assisted therapy in patients with schiz-ophrenia and addiction (dual diagnosis)” by Miguel Monfort, Ana Benito, Gonzalo Haro, Alejandro Fuertes, Monserrat Cañabate and Abel Baquero for consideration by the International Journal of Environmental Research and Public Health.
We confirm that we have followed your instructions, you will be able to check the changes in the forwarded paper, please understand that due to limitations of the study and its context it may not have been possible for us to develop all of them, although we consider that thanks to your invaluable collaboration we have been able to improve our proposal.
We enclose the document in which you kindly helped us, as well as others since we consider that you should have a new global vision with all the modifications. We detail the modifications and detect them in the text:
Revisor 2.
I have reviewed the paper title: "The efficacy of animal-assisted therapy in patients with schizophrenia and addiction (dual diagnosis). The paper is reasonably written that are in line the scope of the journal issue. The relevant and contributes to existing literature. Therefore, I recommend this paper subject to the following the revisions.
- Reorganise the contents of this manuscript to improve the flow and coherence for easy understanding. Where appropriate, insert sub-headings to segregate different points for clear illustration and discussion.
Response 1: This aspect is modified in the text,although the requested modification cannot be modified further, due to the indications of other reviewers.
- In the abstract and introduction, clearly specify the research problems and research objectives. These are important components of a manuscript in guiding readers through the research. What’s the keypoint that significant for Schizophrenia patients for this study design. What's paper to clear research gap? please clarify.
Response 1: This aspect is modified in the text
- Introduction; You should indicate a gap in the previous research, or extend previous knowledge in some way. The introduction is not clear and does not help the reader in the understanding of the whole paper. The topic investigated is not clarified nor explained in terms of the main results obtained. Moreover, please add an outline of the paper at the end of the introductory section that will help the reader in understanding the complex structure of the paper.
Response 1: This aspect is modified in the text
- In the methodology section, expand the methods used in 2 group design, Have the data been validated? how to ascertain the validity of the information used for this research? So, The methodological, material and methods section does not contain any methods at this stage. Please better explain the details regarding the questionnaire (when, where, how are selected firms, which sectors etc.). Please report the psychometric properties of all assessment/questionaires that you used in this research. And brief the detail of your program 10 sessions.
Response 1: This aspect is modified in the text, We have added the criterio and criterio criterio.
- Suggest including a research framework covering all the relevant variables for clear illustration and easy understanding.
Response 1: This aspect is modified in the text
- Include a section on practical implications for possible adoption by the targeted audience. Should there be policy recommendations from this research in mental health and psychiatric fieldwork?
Response 1: This aspect is commented in the discussion.
- Ensure in-text citations are done in the correct format. Please refer to manuscript formatting guidelines.
- Table 1, some symbol not clear, please check it.
Response 1: This aspect is modified in the text
- Figure 2. I think you can upgrade to high figure standard. Please improve design of illustration and Professional styles of figure.
Response 1: This aspect is modified in the text
- The discussion section also need to add compatibility with recent studies and how the results of this studies varied compared to existing studies.
Response 1: This aspect is commented in the discussion.
- The policy recommendations required extension in terms of key variables.
Response 1: This aspect is modified in the text
- How about the limitation of this study? Please explain.
Response 1: This aspect is commented in the discussion.
- The manuscript required native Enlight edit and proofing, The are a few grammatical and sentence structure errors.
Response 1: This aspect is modified in the text
- Please confirm the reference styles, as several references in the bibliography show inconsistent formatting. Some references have the complete name of journals while others have abbreviations. Please update and make it consistent.
Response 1: This aspect is modified in the text
This article is strongly recommended for publication after incorporating certain changes. This article needs thorough proofreading. Overall quality of Language is good. Just minor grammatical mistakes are found. All tables and figures are relevant. Research Methodology has been well defined. All data are aligned to the findings of the research. This article is good attempt in the field experimental research design and will be beneficial for future researchers.
Thank you for your consideration of this manuscript.
Sincerely,
Abel Baquero

Reviewer 3 Report
Major points
- It would be interesting to describe more data about the sample recruitment. Why did not the author present a baseline measurement and only measure from 3rd session?
- If groups were formed by sequential allocation during 24-month recruitment – was there a refusal to participate? Was there dropout in the control group (because the groups had unequal sizes)? Could the ones with the best clinical condition be those who stopped participating in the CG and were discharged from the residential treatment? If they remained in the final sample, would there be no difference between those who received and those who did not receive AAT?
- It would be enlightening if they briefly describe whether they evaluated the characteristics of the data to know if they were allowed the use of parametric analysis. Note that the samples are of very small, unequal sizes, so it would be interesting to inform if there was a correction for multiple comparisons (the authors referred to 3 measurement times and interaction between factors - group and time). In addition, it seemed that in the first measure (Table 1) the scores of the two scales (severity and functionality) had a greater variation (by SD values) in the CG than the EG. (By the way, in case of publication, I think it would be interesting to consult the editing team about improving the layout of the table and whether the data would be better represented in table 1 or in the graphs since they seems to refer to the same data.)
- Still about data and the graphs, I don't know if the authors are using any specific Spanish PANSS version to have a score variation in the positive and negative scales above 49. Anyway, in case the authors decide to keep the graphs, I suggest that at least the scale ranges of the positive and negative dimensions were standardized.
- (Note that, if the authors evaluated that the data can be used in parametric analysis - could the question of groups starting from different points regarding the negative PANSS score be controlled by placing this initial score as a covariate in the model?)
- Although the authors mentioned it as a limitation, the information about the study having a multicenter design is mentioned only in two points of the text - in the abstract (line 18) and in line 77. The line 77 was part of the methodology section, but there is no description of how many centers collaborated in the composition of the sample and how the treatment considered as usual could vary among them. As the reader did not have access to this information, it is not possible to exclude a confusing effect over the result. Did only the EG receive group intervention or could also the CG receive group-treatments? Could the symptom-reducing effect be attributed to contact with group therapists (“social educator, psychologist, AAT technician, and dog trainer”) even if they were not exposed to the therapy dog? Even though I see great therapeutic potential in the use of animals in health assistance - the point is that this therapeutic resource means another cost for any institution, with, I imagine, the demand of specific training. Whether the effect is due to group therapists and not specifically to dogs - this would be a cheaper intervention.
- I completely understand the difficulty of randomizing the allocation in clinical settings, but how much was it possible to preserve the knowledge that the volunteers were being treated differently? This information could certainly influence the patients' reports regarding their symptoms and functionality. And when I cannot blind the volunteers about the treatment, I must blind the evaluators (this is a fundamental point). I do not believe I have read the information described about blinding raters in the manuscript.
Minor points
- A minor suggestion - to be considered considering the guidelines for titles - is to explicitly include the expression "dual diagnosis" in the title and not as an expression in parentheses.
- A revision in favor of the elegance of the text would be interesting, for example, of passages such as "we evaluated to evaluate" (line 197). This is just a matter of writing style, but I believe it will be interesting for the reader.
- I understand that the reader will be able to deduce the use of the colors red and blue from the meaning of the regression in figure 2 (significant and not significant). Even so, it may be interesting to put a legend for the colors since the authors ended up opening a "notes" field to put the meaning of the abbreviation AAT.
Author Response
Abel Baquero Escribano
Department of Medicine and Surgery, Universidad Cardenal Herrera-CEU. Castellón
13Th May 2022
International Journal of Environmental Research and Public Health
Dear Revisor 3:
Thank you for your invaluable help in reviewing our paper entitled “The efficacy of animal-assisted therapy in patients with schiz-ophrenia and addiction (dual diagnosis)” by Miguel Monfort, Ana Benito, Gonzalo Haro, Alejandro Fuertes, Monserrat Cañabate and Abel Baquero for consideration by the International Journal of Environmental Research and Public Health.
We confirm that we have followed your instructions, you will be able to check the changes in the forwarded paper, please understand that due to limitations of the study and its context it may not have been possible for us to develop all of them, although we consider that thanks to your invaluable collaboration we have been able to improve our proposal.
We enclose the document in which you kindly helped us, as well as others since we consider that you should have a new global vision with all the modifications. We detail the modifications and detect them in the text:
It would be interesting to describe more data about the sample recruitment. Why did not the author present a baseline measurement and only measure from 3rd session?
Response 1: Due to problems at the start of the program, this measure could not be taken, although as you indicate it is important to take it into account.
If groups were formed by sequential allocation during 24-month recruitment – was there a refusal to participate? Was there dropout in the control group (because the groups had unequal sizes)? Could the ones with the best clinical condition be those who stopped participating in the CG and were discharged from the residential treatment? If they remained in the final sample, would there be no difference between those who received and those who did not receive AAT?
Response 1: This aspect is modified in the text
It would be enlightening if they briefly describe whether they evaluated the characteristics of the data to know if they were allowed the use of parametric analysis. Note that the samples are of very small, unequal sizes, so it would be interesting to inform if there was a correction for multiple comparisons (the authors referred to 3 measurement times and interaction between factors - group and time). In addition, it seemed that in the first measure (Table 1) the scores of the two scales (severity and functionality) had a greater variation (by SD values) in the CG than the EG.
Response 1: This aspect is modified in the text, checking for normality with the Shapiro-Wilk test and equality of variances with the Levene test. We have added in the limitations: the small sample size and the inequality between the groups would make it convenient to use the non-parametric test, however, the non-parametric analysis of the interaction is not incorporated in the statistical packages , so MANOVA has been used.
We have added in statistical analysis: Bonferroni correction for multiple comparisons was used.
(By the way, in case of publication, I think it would be interesting to consult the editing team about improving the layout of the table and whether the data would be better represented in table 1 or in the graphs since they seems to refer to the same data.)
Response 1: We have included both because the graphics capture the interaction effect better, but of course the editing team decides
Still about data and the graphs, I don't know if the authors are using any specific Spanish PANSS version to have a score variation in the positive and negative scales above 49. Anyway, in case the authors decide to keep the graphs, I suggest that at least the scale ranges of the positive and negative dimensions were standardized.
(Note that, if the authors evaluated that the data can be used in parametric analysis - could the question of groups starting from different points regarding the negative PANSS score be controlled by placing this initial score as a covariate in the model?)
Response 1: Very good observation. To standardize the results, we use the PANSS percentile scores, which we have specified in the text and in the graphs
Although the authors mentioned it as a limitation, the information about the study having a multicenter design is mentioned only in two points of the text - in the abstract (line 18) and in line 77. The line 77 was part of the methodology section, but there is no description of how many centers collaborated in the composition of the sample and how the treatment considered as usual could vary among them. As the reader did not have access to this information, it is not possible to exclude a confusing effect over the result. Did only the EG receive group intervention or could also the CG receive group-treatments? Could the symptom-reducing effect be attributed to contact with group therapists (“social educator, psychologist, AAT technician, and dog trainer”) even if they were not exposed to the therapy dog? Even though I see great therapeutic potential in the use of animals in health assistance - the point is that this therapeutic resource means another cost for any institution, with, I imagine, the demand of specific training. Whether the effect is due to group therapists and not specifically to dogs - this would be a cheaper intervention.
I completely understand the difficulty of randomizing the allocation in clinical settings, but how much was it possible to preserve the knowledge that the volunteers were being treated differently? This information could certainly influence the patients' reports regarding their symptoms and functionality. And when I cannot blind the volunteers about the treatment, I must blind the evaluators (this is a fundamental point). I do not believe I have read the information described about blinding raters in the manuscript.
Response 1: Attempts to conceptualize in the text
Minor points
A minor suggestion - to be considered considering the guidelines for titles - is to explicitly include the expression "dual diagnosis" in the title and not as an expression in parentheses.
Response 1: This aspect is modified in the text
A revision in favor of the elegance of the text would be interesting, for example, of passages such as "we evaluated to evaluate" (line 197). This is just a matter of writing style, but I believe it will be interesting for the reader.
Response 1: This aspect is modified in the text
I understand that the reader will be able to deduce the use of the colors red and blue from the meaning of the regression in figure 2 (significant and not significant). Even so, it may be interesting to put a legend for the colors since the authors ended up opening a "notes" field to put the meaning of the abbreviation AAT.
Response 1: This aspect is modified in the text
Thank you for your consideration of this manuscript.
Sincerely,
Abel Baquero
Reviewer 4 Report
This ms reports the results of a study investigating AAT vs treatment-as-usual for people receiving residential treatment for schizophrenia and substance abuse. This is a timely study, given the increase in AAT in some parts of the world. However, before I can recommend it for publication, I would like to see some things clarified, especially the methods (further detail below).
Abstract
L20 - explain what sort of treatment was received by the control group
L22 - confirm that this was group therapy
keywords - some of the keywords are already in the title, which means they don't need to be in the keywords. The others are very broad in scope. This website may help the authors select appropriate keywords to enhance visibility of this work: https://getproofed.com.au/writing-tips/how-to-pick-the-best-keywords-for-a-journal-article/
Intro
L34 - this phase is unclear, starting with '...difficulty inherent...' until the end of the sentence.
L35 - the NIH definition doesn't seem to be comprehensive. Could it be more clearly defined?
L37 - provide a brief example of a negative symptom. This term is mentioned frequently in the text but never actually defined, until positive symptom, which is described in L39-41.
Materials and methods
L79 - were there any other inclusion criteria screened for? What about interest in, or fear of, dogs?
L87 - much more info about the AAT needs to be provided. Who facilitated the sessions, out of the several people who were involved? on that note, why were so many people involved? what was the role of each person? What was the dog doing during the sessions? Was the dog actively integrated into the therapy, or just present when people needed to be comforted? A major limitation of AAT research generally is that the intervention itself is rarely described. This is an opportunity to help improve the evidence base.
Also, what was treatment as usual? The treatment information for the control group should also be explained clearly.
L92 - how were the PANSS and LSP rated? Was it patient ratings? carer? doctor? What were the scale options and range? What did a higher score represent?
L98 - what descriptive variables?
L99 - what categorical variables?
L103 - suggest changing the mediation analysis to focus solely on Hyp 3. As it is, the figure is very confusing. If the authors focus on Hyp 3, it will simplify the figure.
L114-123 - why are assistance dogs relevant to this study? They perform a different role and have different public access rights. Clarify
L123 - was the ethics approval number for human ethics, animal ethics, or both?
Results
For frequencies - suggest adding n's as well as percentages. With the small sample size, it will be useful to know how many people 2.8% is, for example.
L137 - someone did not complete ANY education? Is that even possible? Or legal? Clarify.
L143 - The baseline schizophrenia symptoms/severity were not provided anywhere in this report, but the authors mention that negative symptoms were different at baseline. Please provide all of the baseline symptom info in the results section, as was done with substance abuse, medications, and physical health conditions.
L161-163 - there is a placeholder for something that is missing in each of the stat test results. It might be the partial eta squared symbol. Please check this.
Table 1 - last two columns: explain that t = time and i = interaction, either in the table itself or in the caption.
Fig 1 - note whether an increase over time = better or worse for each graph. Also suggest amending somewhat so people who print in grey scale can interpret the figures easily.
Fig 2 - as noted above, I suggest changing this figure to reflect only Hyp 3. Also, what do the different coloured arrows represent? This is not clear.
Discussion
L269 - suggest moving this sentence to the bottom of the previous para, and starting the next para with the following sentence, which starts with 'Another study...'
L287 re: limitations. Could it also be that the additional therapy time spent with the patients made a difference, without considering the animal, per se? It's possible that additional treatment of any kind would help improve the outcomes.
L300 - 302 - suggest removing this sentence, since it's impossible to know the impact on negative symptoms due to baseline differences.
Appendix A - why are the tables and figures that are already in the main body of the paper presented as an appendix? That is not necessary. Appendices should be for information that is not presented in the main body of the work, but might be relevant for a small number of people (e.g., who are performing a meta-analysis or intend to replicate the study).
Author Response
Abel Baquero Escribano
Department of Medicine and Surgery, Universidad Cardenal Herrera-CEU. Castellón
13Th May 2022
International Journal of Environmental Research and Public Health
Dear Revisor 4:
Thank you for your invaluable help in reviewing our paper entitled “The efficacy of animal-assisted therapy in patients with schiz-ophrenia and addiction (dual diagnosis)” by Miguel Monfort, Ana Benito, Gonzalo Haro, Alejandro Fuertes, Monserrat Cañabate and Abel Baquero for consideration by the International Journal of Environmental Research and Public Health.
We confirm that we have followed your instructions, you will be able to check the changes in the forwarded paper, please understand that due to limitations of the study and its context it may not have been possible for us to develop all of them, although we consider that thanks to your invaluable collaboration we have been able to improve our proposal.
We enclose the document in which you kindly helped us, as well as others since we consider that you should have a new global vision with all the modifications. We detail the modifications and detect them in the text:
Revisor 4
This ms reports the results of a study investigating AAT vs treatment-as-usual for people receiving residential treatment for schizophrenia and substance abuse. This is a timely study, given the increase in AAT in some parts of the world. However, before I can recommend it for publication, I would like to see some things clarified, especially the methods (further detail below).
Abstract
L20 - explain what sort of treatment was received by the control group
Response 1: This aspect is modified in the text
L22 - confirm that this was group therapy
Response 1: This aspect is modified in the text
keywords - some of the keywords are already in the title, which means they don't need to be in the keywords. The others are very broad in scope. This website may help the authors select appropriate keywords to enhance visibility of this work: https://getproofed.com.au/writing-tips/how-to-pick-the-best-keywords-for-a-journal-article/
Response 1: This aspect is modified in the text
Intro
L34 - this phase is unclear, starting with '...difficulty inherent...' until the end of the sentence.
Response 1: This aspect is modified in the text
L35 - the NIH definition doesn't seem to be comprehensive. Could it be more clearly defined?
Response 1: This aspect is modified in the text
L37 - provide a brief example of a negative symptom. This term is mentioned frequently in the text but never actually defined, until positive symptom, which is described in L39-41.
Response 1: This aspect is modified in the text
Materials and methods
L79 - were there any other inclusion criteria screened for? What about interest in, or fear of, dogs?
Response 1: This aspect is modified in the text
L87 - much more info about the AAT needs to be provided. Who facilitated the sessions, out of the several people who were involved? on that note, why were so many people involved? what was the role of each person? What was the dog doing during the sessions? Was the dog actively integrated into the therapy, or just present when people needed to be comforted? A major limitation of AAT research generally is that the intervention itself is rarely described. This is an opportunity to help improve the evidence base.
Response 1: This aspect is modified in the text
Also, what was treatment as usual? The treatment information for the control group should also be explained clearly.
Response 1: This aspect is modified in the text
L92 - how were the PANSS and LSP rated? Was it patient ratings? carer? doctor? What were the scale options and range? What did a higher score represent?
how were the PANSS and LSP rated? Was it patient ratings? carer? doctor? What were the scale options and range? What did a higher score represent?
L98 - what descriptive variables?
Response 1:they are detailed in results
L99 - what categorical variables?
Response 1:they are detailed in results
L103 - suggest changing the mediation analysis to focus solely on Hyp 3. As it is, the figure is very confusing. If the authors focus on Hyp 3, it will simplify the figure.
Response 1: This aspect is modified in the text
L114-123 - why are assistance dogs relevant to this study? They perform a different role and have different public access rights. Clarify
Response 1: we consider that it is specified in the context of the study, as well as in the legislation reviewed.
L123 - was the ethics approval number for human ethics, animal ethics, or both?
Response 1: for both, the ethics committee assesses all aspects of research, both for humans and animals.
Results
For frequencies - suggest adding n's as well as percentages. With the small sample size, it will be useful to know how many people 2.8% is, for example.
Response 1: We understand it, but we consider that in the context of research and results these aspects are conceptualized.
L137 - someone did not complete ANY education? Is that even possible? Or legal? Clarify.
Response 1:Unfortunately, in the system where the research is carried out, it is something common.
L143 - The baseline schizophrenia symptoms/severity were not provided anywhere in this report, but the authors mention that negative symptoms were different at baseline. Please provide all of the baseline symptom info in the results section, as was done with substance abuse, medications, and physical health conditions.
Response 1:Unfortunately due to problems unrelated to the investigation, it was not possible to collect some data like this.
L161-163 - there is a placeholder for something that is missing in each of the stat test results. It might be the partial eta squared symbol. Please check this.
Response 1: This aspect is modified in the text
Table 1 - last two columns: explain that t = time and i = interaction, either in the table itself or in the caption.
Response 1: This aspect is modified in the text
Fig 1 - note whether an increase over time = better or worse for each graph. Also suggest amending somewhat so people who print in grey scale can interpret the figures easily.
Fig 2 - as noted above, I suggest changing this figure to reflect only Hyp 3. Also, what do the different coloured arrows represent? This is not clear.
Response 1: This aspect is modified in the text, To standardize the results, we use the PANSS percentile scores, which we have specified in the text and in the graphs.
Discussion
L269 - suggest moving this sentence to the bottom of the previous para, and starting the next para with the following sentence, which starts with 'Another study...'
Response 1: we have moved the sentence to the bottom of the previous paragraph.
L287 re: limitations. Could it also be that the additional therapy time spent with the patients made a difference, without considering the animal, per se? It's possible that additional treatment of any kind would help improve the outcomes.
Response 1: we do agree this may be a limitation. We have added: "Also, as the AAT was not compared to another complementary therapy, extra time spent in therapy by the AAT group may have hindered our results"
L300 - 302 - suggest removing this sentence, since it's impossible to know the impact on negative symptoms due to baseline differences.
Response 1: we have removed the sentence.
Appendix A - why are the tables and figures that are already in the main body of the paper presented as an appendix? That is not necessary. Appendices should be for information that is not presented in the main body of the work, but might be relevant for a small number of people (e.g., who are performing a meta-analysis or intend to replicate the study).
Response 1: we try to modify this aspecto in the text
Thank you for your consideration of this manuscript.
Sincerely,
Abel Baquero

Round 2
Reviewer 2 Report
I’m satisfy this revise version.
Author Response
Dear colleague, thanks for your necessary contributions, we appreciate your effort and interest, you will see how we have corrected them in the text.
Thank you.
We all have the problem of not having enough time for so many daily tasks, but the new manuscript version made me think that the authors weren't careful enough in order to consider the suggestions done.
Very kind contribution, we have revised the text, we hope that there are no errors.
Thank you

Reviewer 3 Report
We all have the problem of not having enough time for so many daily tasks, but the new manuscript version made me think that the authors weren't careful enough in order to consider the suggestions done.
Major points
I hope the authors forgive me for the insistence, but I insist on the same considerations as in the previous review. In case of not having enough space, this will require more writing work (the shorter the text, the better). The methodological considerations must be, at least, to be commented as limitations:
- It would be interesting to describe more data about the sample recruitment (sample derived from multicenters). As the reader did not have access to this information, it is not possible to exclude a confusing effect over the result.
- Operational difficulty to register a baseline measurement
- If groups were formed by sequential allocation during 24-month recruitment. There was there a refusal to participate. It was not possible to disentangle effect of the AAT from the clinical characteristics at the dropout moment (in samples of small size, low or high scores would interfere on the mean)
- Authors must review tables and figure 1 before publication (PANSS graphics should be draw at the same scale).
- Authors evaluated that the data can be used in parametric analysis - could the fact of groups starting from different points regarding the negative PANSS score be controlled by placing this initial score as a covariate in the model?
Minor points
- If the cover helps sell a book, does the title help us want to read an article? My suggestion regarding the use of the term "dual diagnosis" was not just to remove the parenthesis. Whether it is an important concept for authors, it should actually be included in the title.
- I imagine that the changes in the text of the introduction occurred in response to the questioning of the other reviewers. Please excuse me, but I liked some excerpts better as they were in the original. In addition, the excerpt on line 49 turned the sentence too long. Would it be more interesting to divide it into two sentences in order to make it easier for any reader?
- If these request does contradict another reviewer's request, leave it as it is. Otherwise, it is necessary to correct the description of the objective in the abstract (on the line 18, “to intervene in the remission of negative symptoms”). I’ve understood that the objective was to see the target in both positive and negative symptoms, not only in negative.
- There are many typos! As just a few examples: (1) with the change done on line 22, the sentence has no closing parentheses; (2) there are duplicate acronyms (“PANSS”) at the same sentence - lines 25 and 26; (3) remove excess period in sentence from line 75 (“resources. [19–21].”); (4) correct typing of the word on line 76 (“Patientes”); (5) “TAA” on line 210; (6) used or wrong verbal tense (“will be used”, line 136); (7) why are there two subtitles with the same text, but identified with different numbers (3.1.1 and 3.1.2 “Sociodemographic characteristics”)? Another reviewer may have asked for some changes here, but the 3.1 subtitle was enough for me...; (8) what does the authors want to mean by “half” in Table 1? Anyway, in a future submission, the authors should do it with greater care (there are many, many errors!)
- In my Ethics committee, we do not consider an exclusion criterion to be simply a statement of an inclusion criterion just written the opposite way. So, it would make the sentence “the presence of a mental disorder other than a schizophrenia spectrum disorder” unnecessary.
Author Response
First of all, we are very sorry that the rush has caused us to be careless in the modifications. We hope we have resolved these oversights now. Thank you very much for your comments.
It would be interesting to describe more data about the sample recruitment (sample derived from multicenters). As the reader did not have access to this information, it is not possible to exclude a confusing effect over the result.
Thank you very much for insisting on this, we are sorry for the confusion: the word multicenter is a typo, we have eliminated it and we have described the residential center where the study was carried out: in residential treatment, specifically, the detoxification program in the Therapeutic Community of Project Amigó Castellón (lines).
Operational difficulty to register a baseline measurement.
Sorry, we replied to you in the coverletter, but we didn't write it in the text. We have added as a limitation: another important limitation is that, due to difficulties with the start of the study, the subjects were not evaluated at baseline, but instead began to be evaluated in the third sesión (lines).
If groups were formed by sequential allocation during 24-month recruitment. There was there a refusal to participate. It was not possible to disentangle effect of the AAT from the clinical characteristics at the dropout moment (in samples of small size, low or high scores would interfere on the mean)
We have statistically verified that there were no differences in the percentage of dropouts and exclusions in both groups and we have added the following limitation: Another important issue is that although exclusions and dropouts occurred equally in both groups, it is not possible to disentangle the effect of AAT from clinical characteristics at the time of dropout or refusal to participate, which may influence outcomes averages as it is a small sample (lines).
Authors must review tables and figure 1 before publication (PANSS graphics should be draw at the same scale).
PANSS graphics have been drawn at the same scale.
Authors evaluated that the data can be used in parametric analysis - could the fact of groups starting from different points regarding the negative PANSS score be controlled by placing this initial score as a covariate in the model
It is a great idea, unfortunately in SPSS’ MANOVA if the variable has been entered as within subjects, it is no longer possible to also enter it as a covariate. We have included this issue in limitations: Another problem with MANOVA is that if a variable has been entered as within subjects, it is not possible to also enter it as a covariate, which would have been very useful with the negative PANSS score, since the two groups start at different points (lines).
Minor points
If the cover helps sell a book, does the title help us want to read an article? My suggestion regarding the use of the term "dual diagnosis" was not just to remove the parenthesis. Whether it is an important concept for authors, it should actually be included in the title.
Response 1: title modified
I imagine that the changes in the text of the introduction occurred in response to the questioning of the other reviewers. Please excuse me, but I liked some excerpts better as they were in the original. In addition, the excerpt on line 49 turned the sentence too long. Would it be more interesting to divide it into two sentences in order to make it easier for any reader?
Reponse 1: sentences divided.
If these request does contradict another reviewer's request, leave it as it is. Otherwise, it is necessary to correct the description of the objective in the abstract (on the line 18, “to intervene in the remission of negative symptoms”). I’ve understood that the objective was to see the target in both positive and negative symptoms, not only in negative.
Response: modified in line 20.
There are many typos! As just a few examples: (1) with the change done on line 22, the sentence has no closing parentheses; (2) there are duplicate acronyms (“PANSS”) at the same sentence - lines 25 and 26; (3) remove excess period in sentence from line 75 (“resources. [19–21].”); (4) correct typing of the word on line 76 (“Patientes”); (5) “TAA” on line 210; (6) used or wrong verbal tense (“will be used”, line 136); (7) why are there two subtitles with the same text, but identified with different numbers (3.1.1 and 3.1.2 “Sociodemographic characteristics”)? Another reviewer may have asked for some changes here, but the 3.1 subtitle was enough for me...; (8) what does the authors want to mean by “half” in Table 1? Anyway, in a future submission, the authors should do it with greater care (there are many, many errors!)
Response: All the changes that you indicate to us have been reviewed and reformulated as well as others.
In my Ethics committee, we do not consider an exclusion criterion to be simply a statement of an inclusion criterion just written the opposite way. So, it would make the sentence “the presence of a mental disorder other than a schizophrenia spectrum disorder” unnecessary.
Very interesting contribution, sorry we did not understand it we have tried to adapt it in the text.
